# The Omnipresence of DYRK1A in Human Diseases

**DOI:** 10.3390/ijms23169355

**Published:** 2022-08-19

**Authors:** Estelle Deboever, Alessandra Fistrovich, Christopher Hulme, Travis Dunckley

**Affiliations:** 1ASU-Banner Neurodegenerative Disease Research Center, Biodesign Institute, Arizona State University, Tempe, AZ 85281, USA; 2Department of Chemistry and Biochemistry, College of Science, The University of Arizona, Tucson, AZ 85721, USA; 3Division of Drug Discovery and Development, Department of Pharmacology and Toxicology, College of Pharmacy, The University of Arizona, Tucson, AZ 85721, USA

**Keywords:** DYRK1A, protein kinase, neurodegenerative diseases, cancers, diabetes, viral infections, heart diseases

## Abstract

The increasing population will challenge healthcare, particularly because the worldwide population has never been older. Therapeutic solutions to age-related disease will be increasingly critical. Kinases are key regulators of human health and represent promising therapeutic targets for novel drug candidates. The dual-specificity tyrosine-regulated kinase (DYRKs) family is of particular interest and, among them, DYRK1A has been implicated ubiquitously in varied human diseases. Herein, we focus on the characteristics of DYRK1A, its regulation and functional role in different human diseases, which leads us to an overview of future research on this protein of promising therapeutic potential.

## 1. Introduction

According to recent estimates, the global population will reach around 9 billion in 2070 [1]. Increasing population will present many healthcare challenges, particularly because the worldwide population has never been older [2]. Recent models have shown that the percentage of people suffering from a disease will double in the coming 20 years, especially for age-related neurodegenerative diseases such as Alzheimer’s disease (AD) and other dementias [2]. Over the past 25 years, technological advances in genomics and analytical approaches have permitted the dissection of the human genome in unprecedented depth, greatly expanding our understanding of many diseases, and facilitating the identification of currently understudied, but potentially druggable, proteins [3,4].

Within the human genome, the kinome accounts for 2% of total protein coding genes and is one of the major superfamilies of homologous proteins with more than 500 members [5,6]. Protein kinases (PK) are responsible for protein phosphorylation by catalyzing the transfer of α-phosphate of ATP to the hydroxyl groups (on serine and threonine for serine/threonine kinases) or phenolic groups (on tyrosine for tyrosine kinases) of proteins to generate phosphate monoesters [5,7]. Protein phosphorylation provides the basis of many biological signaling pathways. Mutations in kinases and the deregulated signaling that results are frequently the cause of many human diseases [6,8,9]. Subsequently, these enzymes have been actively studied around the world as more than 155,000 new references were published in the last five years (July 2022). With 80–90 articles published daily, it is clear for the scientific community that this protein superfamily plays a key role in human health, and recently as therapeutic targets for novel drug candidates [8].

## 2. The DYRK Family

The serine/threonine family of kinases is of particular interest and especially the cyclin-dependent kinases (**C**DKs), mitogen-activated protein kinases (**M**APKs), glycogen synthase kinases (**G**SKs) and CDK-like kinases (**C**LKs), collectively called the **CMGC** family [5]. Within the CMGC family, the human DYRK (dual-specificity tyrosine-regulated kinase) branch comprises five members divided into two classes. DYRK1A is the most-studied member and, along with DYRK1B, forms the class 1 DYRKs. The class 2 DYRK family contains three members: DYRK2, 3 and 4 [10,11]. From yeasts to humans, and recently also discovered in fish, DYRKs are highly conserved, as orthologous genes have been cloned independently in various eukaryotic organisms (Figure 1) [10,12,13].

Indeed, two members of the family have been reported in *C. elegans* (MBK-1 and MBK-2), whereas three members exist in *Drosophila* (*minibrain*, dDyrk2, and dDyrk3). While MBK-1 and *minibrain* are most closely related to mammalian DYRK1A and DYRK1B, dDyrk2 is related to mammalian DYRK4. Finally, MBK-2 and dDyrk3 are related to mammalian DYRK2 and DYRK3.

Conservation among the different mammalian members is strongest within the kinase domain and a sequence just upstream of the kinase domain, named the DYRK homology (DH-box), which is characteristic of the subfamily and contributes to autophosphorylation of a conserved tyrosine within the kinase domain (Tyr321 in DYRK1A) during maturation of the kinase (Figure 2). However, the paralogous class 1 DYRKs share additional homology outside the kinase domain in a nuclear localization signal (NLS) and a PEST motif (a peptide sequence enriched in proline (P), glutamic acid (E), serine (S), and threonine (T)) within the N-terminal and C-terminal regions, respectively [10,11,21,22]. Due to the PEST sequence, these kinases are assumed to have a short half-life [23]. DYRK1A is distinct from all others due to the presence of a unique region containing 13 consecutive histidine residues in the C-terminal region.

DYRK1A, being the most studied member of the DYRK family, has been implicated in a large and growing number of human diseases. Herein, we will focus on the characteristics of DYRK1A, its regulation and functional role in various human diseases. Finally, we will give an overview of future research on this protein of considerable interest.

## 3. DYRK1A Expression and Enzymatic Activity

From kidneys to bone marrow, DYRK1A expression is ubiquitously observed in all human tissues from early embryonic development to adulthood [25,26]. High expression of Dyrk1A was detected in several areas of the adult hindbrain, particularly in the cerebellum and functionally related structures such as the precerebellar and cerebellar nuclei and the vestibular nuclear complex [27]. Subcellular localization of the kinase, however, is still the focus of numerous studies as, in addition to a PEST sequence at its C-terminus (Figure 2), Dyrk1A has a nuclear targeting sequence at its N-terminus. However, studies showed that a substantial amount of endogenous Dyrk1A is localized within the cytoplasm of the cells in the brains of humans, mice, and chickens [28,29]. This dual localization is due to the fact that DYRK1A substrates are both nuclear and cytosolic proteins [10,29]. While DYRK1A is mainly localized in the cytosol in endogenous conditions, it is also accumulated in the nucleus when exogenously overexpressed from substrates that shuttle to and from the nucleus [29,30,31,32].

Like most other kinases, DYRK1A depends on a molecular switch to adopt an active/inactive conformation. Whereas the dual-phosphorylation of the MAPKs is a classical paradigm for the on⁄off regulation by upstream protein kinases, DYRKs rely on autophosphorylation of an absolutely conserved tyrosine (Y321 for DYRK1A) residue in the activation loop (see Figure 2) and appears to be an evolutionarily conserved ancestral feature [33]. It was shown that, for mammalian DYRK1A, tyrosine autophosphorylation is an intrinsic capacity of the catalytic domain and does not depend on other domains or any cofactor. However, Y321 substitution with phenylalanine significantly reduces, at least in vitro, the catalytic activity of DYRK1A and, surprisingly, dephosphorylation does not inactivate mature DYRK1A [22,34]. It is therefore assumed that tyrosine phosphorylation is only required for activation but not for maintenance of the active state.

Based on these findings, DYRK1A has multiple biological functions: signaling, mRNA splicing, chromatin transcription, DNA damage repair, cell survival, cell cycle control, neuronal development and functions, synaptic plasticity, etc. [10,11,12,19,35]. The level of activity of DYRK1A is of key importance for its physiological and pathological effects [24,36,37]. Hence, DYRK1A is regulated based on both gene expression and protein abundance (for reviews see [34,35,36,38]) with both low and high expression exerting significant effects on human diseases (Table 1). Due to its low tissue specificity, multiple proteins are targeted by DYRK1A overexpression, leading to a multitude of various changes within the cell and, consequently, to a plethora of different symptoms and pathologies.

## 4. DYRK1A and Neurological Diseases

Due to its involvement in a multitude of neural development and neuronal signaling pathways, DYRK1A activity has received increased interest as a key contributing factor to neurodevelopmental and neurodegenerative diseases. Deregulated DYRK1A activity has been linked to Down syndrome (DS), Intellectual Developmental Disorder Autosomal Dominant 7 (MRD7), dementia, AD, Fronto-Temporal Degeneration (FTD), Huntington’s disease (HD) and Parkinson’s disease (PD) [82].

### 4.1. Down Syndrome (DS) and Intellectual Developmental Disorder Autosomal Dominant 7 (MRD7)

With an incidence of 1 in 800 lives births, DS, which is caused by a trisomy of chromosome 21, is one of the leading causes of intellectual disability. In addition to all the problems of cognitive development, patients with DS face various health issues throughout their lives, including learning and memory deficits, congenital heart disease, AD, leukemia, and cancer, leading to huge medical and social costs [83].

The DYRK1A gene is located within human chromosome 21q22.2, also known as the Down Syndrome Critical Region (DSCR). There is considerable genetic and pharmacologic evidence showing that the mere 1.5-fold overexpression of DYRK1A is responsible for a variety of symptoms observed in DS patients. In the same way, patients with DYRK1A haploinsufficiency syndrome, also known as MRD7, experience a heterozygous loss-of-function in DYRK1A. This syndrome is characterized by severe intellectual disability, speech and motor delay, autism spectrum disorder and sometimes epileptic seizures (for complete reviews see [84,85,86]). Both DS and MRD7 syndromes affect similar brain areas and functions; but MRD7 is distinguished by the observation of pronounced microcephalies [82,87]. MRD7 is a monogenic disease whereas DS is a polygenic disease and not all the DS clinical phenotypes are assigned to dysregulation of DYRK1A. However, these diseases demonstrate the critical dosage sensitivity of DYRK1A as either too much or too little kinase activity that has negative health consequences [83]. This is likely due to the critical role of DYRK1A in neural development and timing of neural progenitor cell differentiation. Indeed, Najas and co-workers showed, in the Ts65Dn mouse model, that a 1.5-fold increase in DYRK1A protein levels in neural stem cells of the developing cerebral cortex caused by trisomy of the DYRK1A gene lengthens the cell cycle and decreases the production of neurons [88].

As mentioned previously, DYRK1A acts on a multitude of exogenous protein substrates, including translation and transcription factors, splicing factors, miscellaneous proteins or cytoskeletal targets and synaptic proteins (see Table 1 for details) [88,89]. For instance, the levels of Cyclin D1 and DYRK1A proteins during the early phase of cortical neurogenesis are inversely correlated, as DYRK1 induces proteolytic degradation of Cyclin D1 by its phosphorylation at Thr286, thereby altering the careful balance between stem cell division and neural differentiation required for production of a healthy number of functional neurons. The deficit in neuron numbers in the trisomic Ts65Dn mouse is restored by normalizing DYRK1A gene dosage [88]. Additionally, the neural retina is thicker in DS individuals than in the normal population. Increased retinal size and cellularity in Ts65Dn mice correlated with abnormal retinal function and resulted from an impaired caspase-9-mediated apoptosis during development. It was shown that normalization of DYRK1A gene copy number in Ts65Dn mice rescues both morphological and functional phenotypes [47,48]. These observations suggest that DYRK1A inhibition could have some therapeutic value for DS patients. To date, no such parallel has been identified in MRD7 patients, although it has been shown in heterozygous DYRK1A +/− mice that such a reduction in protein expression was systematically associated with significant reduction in body weight and length and pronounced microcephaly, thus mirroring some neurological traits associated with the human MRD7 pathology, such as defective social interactions, stereotypic behaviors and epileptic activity, with also altered proportions of excitatory and inhibitory neocortical neurons and synapses [86,90,91].

### 4.2. Dementia and Alzheimer’s Disease (AD)

Approximately 52 million people worldwide had dementia in 2020. Because of the age-related incidence of dementia and the aging population, this number is predicted to almost double every 20 years. AD is the most common cause of dementia and accounts for 60–70% of all cases, representing one of the leading socioeconomic problems in healthcare [2]. The onset and the progression of this neurodegenerative disease are associated notably with amyloid-β peptide (Aβ) aggregation and tau protein hyperphosphorylation [92].

From this perspective, recent evidence implicates DYRK1A as a promising target for AD as it has been associated with both tau and amyloid neuropathologies through DYRK1A-mediated phosphorylation of key substrates: tau, amyloid precursor protein (APP), Neprilysin (NEP), and Presenilin 1 (PSEN1) and SEPT4 (Table 1). A recent work highlights DYRK1A as a modulator of the axonal transport machinery driving APP intracellular distribution in neurons. Indeed, DYRK1A inhibition affects the expression (transcription, translation and/or accumulation) of proteins that are involved in the intracellular trafficking of APP-loaded vesicles [93]. Consistent with this, inhibition of the kinase in AD transgenic mouse models decreased APP and amyloid-β accumulation [94,95,96]. However, in the Dp3Tyb 528 model, which contains an additional copy of DYRK1A, no increase in APP abundance or sign of amyloid-β accumulation was observed [97], suggesting that having three copies of DYRK1A gene is not sufficient to modulate APP protein abundance or promote amyloid-β accumulation.

In 2007, Kimura et al. showed in HEK293 cells that the amount of tau phosphorylated at Thr212 increased by co-transfection of the DYRK1A expression vector, whereas tau level was similar [78]. Tau phosphorylation at Thr212 is associated with Aβ overproduction. Moreover, it was shown that this tau phosphorylation at Thr212 primes tau for phosphorylation by glycogen synthase kinase-3 beta (GSK3-β) at Ser208, contributing to the formation of paired helical filaments composed of highly phosphorylated tau, a component of neurofibrillary tangles (NFTs) [78,98]. In this context, in vivo studies were conducted in various mouse models and all compounds tested were able to reduce tau phosphorylation through inhibition of DYRK1A activity, leading to inhibition of tau oligomerization and aggregation [99,100].

Furthermore, as the adult human brain expresses six isoforms of tau by alternative splicing of its pre-messenger ribonucleic acid (pre-mRNA), which contains 16 exons, splicing modulation by DYRK1A has been observed in various studies [32,39,43,101]. Exclusion or inclusion of tau exon 10 (E10), which encodes the second microtubule-binding repeat, gives rise to tau isoforms with three (3R) or four (4R) microtubule-binding repeats, respectively. In the normal adult human brain, approximately equal levels of 3R-tau and 4R-tau are expressed [39]. Tau E10 splicing is regulated by several serine/arginine-rich proteins (SR) or SR-like proteins, including ASF, SC35, 9G8, Tra2, and SRp55 (Table 1). Alternative splicing factor (ASF) was found to be the most effective tau E10 splicing factor when compared with other SR proteins as it binds to a polypurine enhancer on tau E10 and plays essential and regulatory roles in tau E10 inclusion. By phosphorylating ASF at Ser227, Ser234 and Ser238, DYRK1A transforms its nuclear distribution, making it unavailable to the tau transcript and favoring the production of the 3R-Tau isoform (characteristic for AD and other tauopathies) over the 4R-Tau isoform, creating a strong disequilibrium and therefore neurofibrillary degeneration [43]. It was shown that DYRK1A overexpression might also phosphorylate PS1, 9G8, SC35 and, SRp55 (at Ser280, Ser303, and Ser316) promoting tau exon 10 inclusion and leading to an increase in 3R-tau expression, which may initiate or accelerate tau pathology in AD patients’ brains [11,32,39,43,70,74,78,82,101].

On the other hand, tau dephosphorylation has been observed resulting from increases of both DYRK1A and regulator of calcineurin 1 (RCAN1) expression in human DS and AD brains. Calcineurin is a calcium/calmodulin dependent serine/threonine phosphatase which promotes, by dephosphorylating the nuclear factor of activated T cells (NFAT) transcription factor, NFAT translocation into the nucleus contributing to several genes’ transcription and subsequent events (e.g., cell proliferation, apoptosis, angiogenesis, synaptic plasticity, immune response, and skeletal/cardiac muscle development) [75]. However, RCAN1 overexpression results from the inhibition of signaling pathways that are controlled by NFAT (Table 1) which was shown to directly interact with DYRK1A. Recently, DYRK1A was demonstrated to directly interact with and phosphorylate RCAN1 at Ser112 and Thr192 residues, which would prime the protein for further phosphorylation by other protein targets like GSK3-β at Ser108 [76]. This synergistic interaction of DYRK1A, NFAT and RCAN1 could contribute to a variety of pathological features of DS, including early onset of AD [75,76].

Thus, these findings provide novel insights into the molecular mechanisms of AD and our understanding of neurodegeneration caused by dysregulation of tau and other proteins. Targeting DYRK1A appears to be a doable treatment approach for AD and dementia.

### 4.3. Parkinson’s (PD) and Huntington’s (HD) Diseases and Fronto-Temporal Degeneration (FTD)

PD is the second most common neurodegenerative disease after AD and affects approximately 1% of the population older than 60 years. Dozens of PD-related symptoms exist, both neurological and physical. The most typical features include resting tremor, bradykinesia, rigidity and postural instability, olfactory dysfunction, cognitive impairment, psychiatric symptoms, and autonomic dysfunction. Thus, various PD phenotypes have been highlighted with different pathogenic mechanisms and progression [102]. Most PD cases are idiopathic but 5–10% develop the genetic forms and risk generally increases with the accumulation of environmental exposures (e.g., pesticides, solvents, metals) associated with industrialization [102].

On the contrary, HD and FTD are caused by expansion of simple repeats dominantly inherited CAG in exon 1 of the *huntingtin* gene and mutations in microtubule-associated protein tau (MAPT), progranulin (PGRN), and chromosome 9 open reading frame 72 (C9orf72) expansion mutations, respectively [103,104]. While HD is characterized by progressive involuntary choreiform movements, behavioral and psychiatric disturbances, and dementia, or FTD describes a cluster of neurocognitive syndromes that present with impairment of executive functioning, changes in behavior, and a decrease in language proficiency [105].

Recently, it has been demonstrated that increased DYRK1A levels were present in the brains of patients not only with AD but as well as other neurodegenerative diseases such as PD, HD and FTD syndromes [106,107]. Indeed, DYRK1A-mediated phosphorylation of Parkin (Table 1) inhibits its E3 ubiquitin ligase activity and, consequently, impairs its neuroprotective function in dopaminergic neurons [73]. Likewise, SEPT4, which has been found in neurofibrillary tangles and in α-synuclein-positive cytoplasmic inclusions (Lewy bodies) in PD brains, is phosphorylated by DYRK1A (Table 1) which leads to more α-synuclein (α-syn) aggregation and loss of dopaminergic neurons [77]. This phenomenon is accentuated by the fact that DYRK1A can also directly phosphorylate α-syn, contributing to its aggregation and, potentially, to disease progression [81]. This time, in vivo assays demonstrated that Dyrk1a expression is crucial for the survival of dopaminergic neurons (DA) in the MPTP-induced mouse model of PD due to an abnormal activity of the mitochondrial caspase9 (Casp9)-dependent apoptotic pathway during the main wave of programmed cell death that affects these neurons [108]. Another study, performed on the same PD mouse model, suggested that an increased level of miR-204 results in the death of DA by upregulating the expression of DYRK1A and targeting the DYRK1A-mediated apoptotic signaling pathway [109].

In addition, several specific mutations in the tau gene associated with frontotemporal dementia with Parkinsonism linked to chromosome 17 (FTDP-17) lead to dysregulation of tau E10 splicing and result in a selective increase in either 3R-tau or 4R-tau, which is also related to other neurodegenerative disorders, such as FTD and FTDP-17. It was demonstrated that 9G8 directly interacts with the proximal downstream intron of E10, a clustering region of FTDP mutations, and inhibits tau E10 inclusion, and that DYRK1A interacted with and phosphorylated 9G8 in vitro and in live cells (Table 1). As mentioned previously for AD, this phosphorylation regulates 9G8’s activity and, consequently, tau E10 splicing and 3R-tau and 4R-tau disequilibrium [39]. The equivalent can be stated for SRp55 (Table 1) [32]. Thus, the imbalance of 3R-tau and 4R-tau contributes to several types of neurodegenerative diseases, such as FTDP-17, DS and PD. Compounds that could affect this splicing event, such as Dyrk1a inhibitors, could have significant therapeutic potential.

## 5. DYRK1A and Other Diseases

In addition to neurological disorders, DYRK1A dysregulation has been linked to cancer (especially lung and pancreatic cancers), glioblastoma, melanoma, leukemia, diabetes, and heart diseases [78,110,111]. Consequently, individuals with DS are at increased risk for developing multiple congenital disorders [24].

### 5.1. Diabetes

Diabetes is a group of diseases defined by high levels of blood glucose and is classified in two major subgroups, type 1 diabetes (T1D) and type 2 diabetes (T2D). T1D is also known as the insulin-dependent diabetes or juvenile diabetes, and results from autoimmune destruction of insulin-producing pancreatic β-cells, resulting from genetic and environmental factors, which leads to an insulin production deficiency. T1D patients need endogenous insulin administration to survive. Whereas people with T2D produce insulin, but do not respond effectively to the endogenously expressed insulin, making them “insulin-resistant” [112,113].

There are currently no commercially available drugs able to induce human β cells to replicate and regenerate. However, numerous biological targets have been studied in this context, including DYRK1A. Studies show that DYRK1A small molecule inhibitors induce human β-cell proliferation both in vitro and in vivo [54,78,114,115,116,117,118]. Several studies have demonstrated that DYRK1A overexpression attenuated β-cell proliferation through NFAT dysregulation, a transcription factor that transactivates cell cycle-activating genes and represses cell cycle inhibitor genes including other CMGC, cyclins and p57 (Table 1) [30,68,114,115]. Thus, inhibition of DYRK1A activity could provide a therapeutic avenue for enhancing β-cell numbers in T1D, thereby restoring endogenous insulin production. Promisingly, harmine, a DYRK1A inhibitor and glucagon-like peptide 1 (GLP-1) receptor agonist, has recently been shown to work synergistically to activate proliferation of human pancreatic β-cells in human cadaveric islets ex vivo [114]. Some promising results were also obtained with harmine derivatives and various other DYRK1A inhibitors [67,116]. A recent study revealed the importance of the DREAM complex in enforcing quiescence in adult human β cells and demonstrated that small molecule DYRK1A inhibitors induce human β cells to replicate by converting the repressive DREAM complex into an alternate pro-proliferative configuration containing the MuvB complex and B-MYB (MYBL2) referred to as the “MMB complex” [110]. Although encouraging, the community still awaits clinical entry of a DYRK1A inhibitor for T1D.

### 5.2. Solid Cancers and Leukemias

Development of malignant cells through aberrant gene function and altered gene expression patterns has been the focus of medical research in recent years while dissecting the origins of cancers and leukemias. Growing evidence shows that epigenetic factors are primarily involved in causing these abnormalities, along with genetic alterations [119,120]. Considering the increasing number of emerging tumors (lymphomas, retinoblastomas, etc.) in patients with DS, researchers have investigated the possible link between this genetic disorder and the development of these malignant cells [64]. It appears that the DYRK1A protein again plays a major role in the manifestation of these diseases as its modulation in sarcoma, and lung, pancreatic, and ovarian cancer may result in dysregulation of cell cycle control [45,80].

However, despite numerous research reports in recent years, the role of DYRK1A in cancer and leukemia, in the context of DS, is still very unclear as both oncogenic and tumor suppressive roles have been reported [24]. For example, Recasens and colleagues recently revealed by phosphoproteomics in the context of glioblastoma that DYRK1A functions as a tumor suppressor. Its overexpression deactivates another important kinase, CDK1, which positively regulates mitosis and inhibits the cell cycle (Table 1) [49]. On the other hand, DYRK1A was highly expressed in lung and pancreatic cancer cells, and that its protein level was positively correlated with that of STAT3, c-MET and EGFR (Table 1) as it was found that DYRK1A siRNA could suppress the levels of EGFR and MET receptor tyrosine kinases [46,56,57,58]. In addition, phosphorylation of chromatin-bound transcriptional regulator heterochromatin protein 1 (HP1) (Table 1) by DYRK1A antagonizes HP1-mediated transcriptional repression and participates in abnormal activation of cytokine genes in DS-associated megakaryoblastic leukemia [65]. Moreover, this kinase mediates the DREAM complex assembly (Table 1) leading to ovarian cancer dormancy as DYRK1A inhibition reduced spheroid viability and restored sensitivity to chemotherapy and targeting actively proliferating cells [53].

RCAN1, associated in the development of DS and AD, is also involved in cancer. DS and AD patients develop different forms of cancer compared to controls, including lower incidence of some types of solid cancer and higher incidence of leukemia. RCAN1 plays a bivalent role in cancer by promoting or attenuating cell apoptosis, inhibiting or promoting cell proliferation and migration, and suppressing or facilitating angiogenesis. In addition, RCAN1 targets NFAT which is also activated in several types of cancer [75]. Taken together, these studies show that DYRK1A is an important kinase in multiple cancers [121]. However, its specific functional role appears to be cell type- and context-dependent. While DS individuals have increased risk of developing certain type of cancers, particularly leukemia, they showed reduced risk for many other types of solid tumors [122,123]. Appreciating these nuances of DYRK1A activity will be critical for the future pursuit of DYRK1A focused cancer therapeutics.

### 5.3. Viral Infections

The ability of viruses to hijack the cellular machinery makes them strategic players in the development of human diseases [68,124]. Several studies have demonstrated interactions between viruses and kinases from the CMGC group. Hence, viruses appear to be able to co-opt these kinases to optimize their replication in the cell and the action of some kinases appears to inhibit viral activity. Within this context, DYRK1A is of significant interest [124]. Indeed, this enzyme phosphorylates SR proteins (Table 1) which play major roles in the regulation of gene expression by controlling constitutive and alternative splicing and thus viral expression. In the context of HIV-1t, DYRK1A regulates the activity of several transcription factors like NFAT and cyclin L2 (Table 1) thus making viruses latent and preventing replication in macrophages [68,124,125]. Moreover, DYRK1A interacts also with oncoproteins from adenovirus and human papillomavirus (HPV) [52,126]. For example, DYRK1A was shown to phosphorylate in vitro and in vivo Thr5 and Thr7 of HPV16E7, which is a high-risk tumorigenic viral protein identified as one of the causative agents for the development of cervical cancer [126]. Finally, a role of DYRK1A gene in the resistance against the severe acute respiratory syndrome coronavirus 2 (SARS-CoV-2), the causative agent of coronavirus disease 2019 (COVID-19), has been recently suggested [127].

### 5.4. Heart Diseases

Whether the origin of the problem is genetic, related to a congenital disease (diabetes, etc.) or to behavior (smoking, nutrition, etc.), the incidence of heart disease continues to be the focus of medical issues all over the world [128]. In search for the cause, kinases, including DYRK1A, have been pointed out because of their interaction with all members of the cyclin D family. In vitro and in vivo studies have demonstrated that overexpression of DYRK1A leads to increased phosphorylation of Ccnd2 (Table 1)**,** promotes its subsequent proteasomal degradation and thus dysregulation, leading to compromised cardiomyocyte proliferation and finally cardiomyopathy [129]. On the other hand, the downregulation of DYRK1A impacts the NFAT transcription factor, which induces cardiomyocytes and cardiac hypertrophy [69]. Taken together, these results suggest that the expression of DYRK1A could contribute to these syndromes due to a DYRK1A role in cardiomyocyte differentiation *vs.* proliferation similar to the effect on neuronal differentiation and proliferation in the brain of DS and MRD7 individuals. Again, individuals with DS are more likely to develop congenital heart defects or cardiac malformations [24]. Future research will show whether downregulation or overexpression of Dyrk1a is a beneficial tool in the treatment of cardiomyopathies as excess Dyrk1a appears to be useful in preventing maladaptive remodeling due to cardiac hypertrophy. More importantly, targeted repression of cardiac Dyrk1a has the potential to promote myocyte cell cycle progression and promote regeneration after myocardial infarction.

## 6. DYRK1A, a Target for the Patient’s Future

Protein kinases represent very attractive and challenging drug targets for industry and academia to tackle complex disorders. In neuroscience, the development of such kinase-targeted therapies has not been primarily investigated due to several issues, including the multifactorial nature of the central nervous system (CNS) diseases, the failure of many advanced CNS clinical trials, the high conservation of the kinase active site, and the lengthy approval process of a novel CNS drug by the Food and Drug Administration [111].

In this context, the case of AD and other dementias is particularly interesting because no effective treatment has yet emerged despite the growing number of candidates and studies on the subject. One reason is that targeting AD pathology at single components is not likely to be possible and will require a multifactorial intervention. In this regard, the inhibition of DYRK1A activity may synergistically improve cognition and reduce the formation of Aβ oligomers through inhibition of APP phosphorylation, while also reducing AD-associated tau hyperphosphorylation and pathology through inhibition of tau phosphorylation and regulation of tau E10 inclusion. Thus, a single target, DYRK1A, would simultaneously attenuate the two major pathologies of AD [94].

For the last 20 years, many studies have demonstrated that DYRK1A dosage and activity play a key role across a variety of human diseases (Figure 3) [37,114,130,131,132,133,134,135]. Over the past decade, the development of novel treatments to regulate DYRK1A activity in cell models and mouse models has exploded. Nevertheless, while many small molecule therapeutics show significant promise in these models, none have yet advanced beyond early clinical trials.

Most DYRK1A-targeted approaches reduce the activity of DYRK1A as almost all pathologies identified to date are due to its overexpression. Consequently, several molecules have been isolated from natural sources and identified as potent in vitro inhibitors, such as harmine and analogues, epigallocatechin gallate (EGCG) and other flavan-3-ols, benzocoumarin, staurosporine, etc. Many of them had significant side effects due to lack of selectivity, so analogues were developed to limit the undesirable effects [132,135]. In addition, synthetic DYRK1A inhibitors have emerged, such as leucettine L41, INDY, FINDY, Dyr219, ALGERNON, etc. [24,37,95,99,100,111,114,130,131,132,134,136]. Most of them are acting as traditional Type I ATP-competitive inhibitors and appear to have significant effects on in vitro and in vivo models of most of the above-mentioned disorders. The main problem for most compounds remains the lack of selectivity, the poor bioavailability and the numerous side effects that occur with those compounds. Also, to date, no study has successfully achieved the opposite phenomenon (i.e., increasing DYRK1A activity). Reduced DYRK1A activity is the main cause of MRD7. Increasing DYRK1A activity would be a potential strategy to alleviate the severe cognitive deficits present in MRD7 and by extrapolation, other pathologies with an autistic spectrum. Nevertheless, no DYRK1A compound activator has been described to date.

Another possible way to achieve reductions to DYRK1A activity is by targeting DYRK1A expression through microRNAs (miRNAs). Indeed, those endogenous non-coding small RNA molecules have emerged as efficacious therapeutic candidates for the management of cancer and other diseases, as they were shown to be involved in many of them [137,138]. Most miRNAs were found to downregulate DYRK1A [109,139,140]. Unfortunately, due to the large number of possible targets, also known as the “too many targets for miRNA effect”, DYRK1A targeted miRNA induces excessive toxicity and side effects [141]. Consequently, proteolysis targeting chimeras (PROTACs) have appeared in drug discovery as safer potential therapeutics as their ligand is designed to bind to a specific protein of interest [142]. Although effective in other cases, notably against breast and prostate cancers, this avenue has not yet been exploited in the case of DYRK1A regulation Broad therapeutic implications related to DYRK1A’s roles as a tumor suppressor and mediator of radiosensitivity were highlighted in the past years [137,143].

## 7. Concluding Remarks and Future Directions

While the role of DYRK1A in cellular and neuronal development has been demonstrated for many years, the realization of its involvement in varied human diseases is much more recent. Its key role across a variety of human diseases makes it a prime target for therapeutic development, particularly because the impact of DYRK1A usually results from its overexpression. However, there is still a long way to go to fully understand its cellular and physiological functions. While the first compounds are under (pre-)clinical trials, many others continue to emerge that seek to improve selectivity and the overall safety profile. Finally, it cannot be excluded that other diseases may yet prove to be dependent on this key kinase. A particularly topical case would be that of viral infections, given the future emergence of recurrent pandemics such as that of COVID-19.

## Figures and Tables

**Figure 1 ijms-23-09355-f001:**
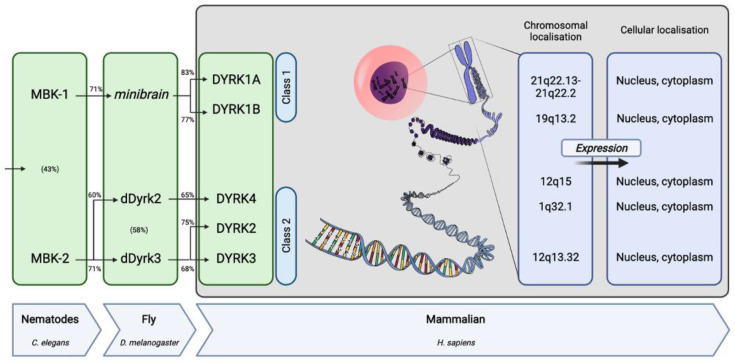
DYRK family of proteins origins and specificities in humans (based on [10,14,15,16,17,18,19,20]). The percentage of conservation at the protein level between orthologues is indicated above the arrows and between two paralogues is indicated in parentheses within the boxes.

**Figure 2 ijms-23-09355-f002:**
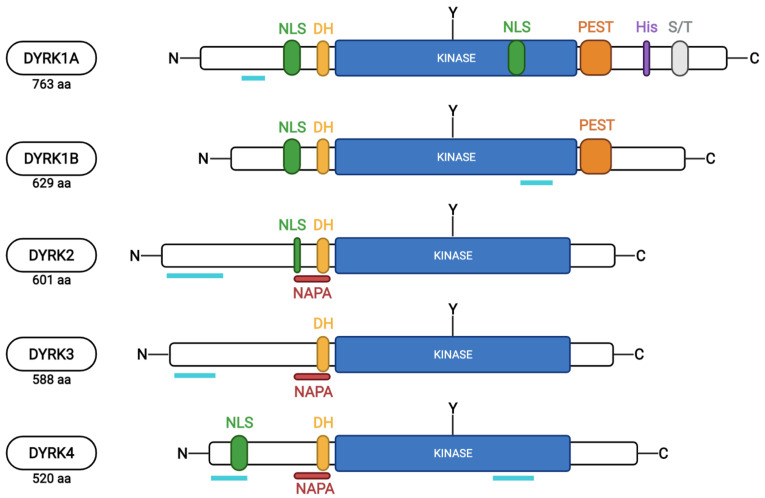
Schematic representation of the protein sequences of the 5 human DYRKs (from uniport.org and based on [10,24]). The different protein motifs identified are indicated: NLS, nuclear localization signal; DH, DYRK-homology box; NAPA, N-terminal autophosphorylation accessory region; kinase, kinase domain; PEST, motif rich in proline, glutamic acid, serine, and threonine residues; His, polyhistidine domain; S/T, serine and threonine-enriched domain; and Y, tyrosine residue autophosphorylated by DYRKs within the activation loop. Turquoise lines indicate protein regions affected by alternative splicing events.

**Figure 3 ijms-23-09355-f003:**
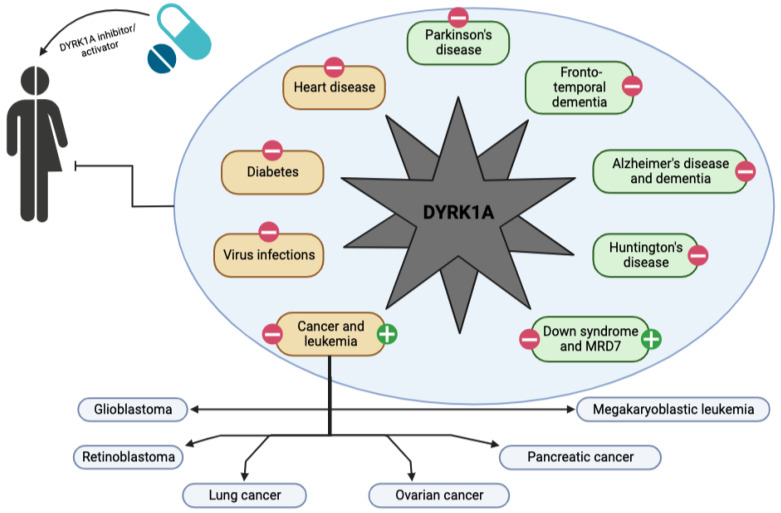
Schematic representation of the spectrum of action of a DYRK1A activator or inhibitor on the development of neurological (in green) and other diseases (in brown) in humans. Pathologies are labeled with a green cross if DYRK1A activation is a potential therapeutic solution and with a red line if DYRK1A inhibition is a potential solution.

**Table 1 ijms-23-09355-t001:** Proteins targeted by DYRK1A, associated or potentially associated with human diseases and acting as interactor (I) and/or substrate (S).

Symbols	Full Names	Functions	Effects	Human Diseases	References
SRDF7/9G8	/	I, S	Inhibition of Tau exon 10 inclusion promotion, imbalance of 3R-tau and 4R-tau expression and neurofibrillary degeneration	Down syndrome, Alzheimer’s and Parkinson’s disease, Fronto-temporal dementia with Parkinsonism linked to chromosome 17	[39]
AMPH	Amphisin	I, S	Depolarization and polarization of isolated synaptosomes and regulation of general neuronal plasticity	Down syndrome (potential)	[40]
APP	Amyloid precursor protein	I, S	Accumulation of β- amyloid peptides (Aβ) in plaques, interference with nMDA receptor function, abnormal calcium influx and neuronal oxidative stress and activation of GSK-3β	Down syndrome, Alzheimer’s disease	[41]
RAD54L2/Arip4	Androgen Receptor Interacting Protein 4	I	Changes in the homeostasis of steroid hormone-controlled cellular events	Down syndrome (potential)	[42]
SRSF1/ASF	Alternative splicing factor	I, S	Increase in 3R-tau level, tau hyperphosphorylation and aggregation in neurofibrillary	Down syndrome, Alzheimer’s, and Parkinson’s disease	[43]
B-Raf	Braf transforming gene	I	Inhibition of neuroprogenitor cells proliferation and premature differentiation	Down syndrome (potential)	[44]
c-MET	/	I	Pancreatic malignant cell proliferation	Pancreatic ductal adenocarcinoma, lung cancer	[45,46]
CASP-9	Cystein aspartyl protease Caspase 9	I, S	Increased retinal size and abnormal retinal function, apoptosis	Down syndrome	[47,48]
CDC23	/	I, S	Degradation of cyclin B, deactivation of CDK1 and retinoblastoma cell proliferation	Glioblastoma	[49]
CREB1	cAMP response element-binding protein	I, S	Inhibition of hippocampal progenitor cells differentiation	Down syndrome (potential)	[46]
Ccnd 1,2 & 3 Ccln L2	Cyclin D1, D2 and D3 Cyclin L2	I, S	Cardiomyocyte proliferation and premature differentiation, inhibition of transcription factors and arresting cell cycle Phosphorylation of Cyclin L2 and cellular degradation	Cardiomyopathy and heart failure associated with Down syndrome, cancer (tumorsupressor) Viral infection (HIV-1)	[24,50]
Dcaf7	DDB1 and CUL4 associated factor 7	I	Huntingtin-associated protein 1 association reduction and growth retardation Mediation of the interaction of the adenovirus E1A oncoprotein	Down syndrome (potential), viral infection	[51,52]
DREAM complex	/	I	Transcription inhibition of cell cycle genes in the G0/G1 phase Proliferation of human pancreatic beta cells	Down syndrome (potential), ovarian cancer, diabetes	[53,54,55]
EGFR	Epidermal growth factor receptor	I	Malignant cell proliferation	Lung cancer, glioblastoma	[56,57]
FOXO1	Forkhead transcription factor FKH R	I, S	Disruption of DNA damage, ROS regulation and cell death in leukemic B cells	Leukemia	[58,59]
GLI1	Glioma-associated oncogene 1	I, S	Cell growth promotion, differentiation, and tissue patterning	Cancer (oncogene)	[60]
GluN2A	Glutamate receptor, ionotropic, NMDA2A	I, S	Synaptic alteration	Down syndrome	[61]
GSK-3β	Glycogen synthase kinase 3beta	I, S	Downregulation of Nrf2, disequilibrium between cellular oxidants and the antioxidative processes, phosphorylation of α-synuclein and adipogenic proteins expression reduction	Alzheimer’s and Parkinson’s disease, obesity (potential)	[62]
HAP1	Huntingtin interacting protein 1	I	Dcaf7 association reduction and hypothalamus growth retardation, neuronal differentiation inhibition and cell death	Down syndrome (potential)	[51,63]
HP1	Heterochromatin protein 1	I, S	Repression of HP-mediated transcription and abnormal activation of cytokine genes	Down syndrome-associated megakaryoblastic leukemia	[64,65]
ID2	/	I, S	Destabilization of transcription factors, loss of gliomna stemness and inhibition of tumour growth	Cancer and glioblastoma	[66]
MEK	Dual specificity mitogen-activated protein kinase	I	Inhibition of neuroprogenitor cells proliferation and premature differentiation	Down syndrome (potential)	[44]
NAFTc	Nuclear factor of activated T cells	I	Angiogenesis promotion, neuroprogenitor cells proliferation inhibition and β-cell proliferation attenuation	Diabetes, heart diseases, cancer (oncogene), viral infection and Down Syndrome	[30,67,68,69]
MME/NEP	Neprilysin	I	Accumulation of β- amyloid peptides (Aβ) in plaques	Alzheimer’s disease	[70]
Notch	Notch Signaling Pathway	I, S	Neural cells signaling attenuation	Down syndrome (potential)	[31]
NRSF/REST	RE1-silencing transcription factor	I	Inhibition of cells proliferation and differentiation	Cancer (tumorsupressor), MRD7 (potential)	[71]
CDKN1B/P27	/	I, S	Inhibition of neuroprogenitor cells proliferation and premature differentiation	Down syndrome (potential), cancer (tumorsupressor)	[50]
P53	Transformation related protein53	I, S	Neuronal proliferation, increase of p21 expression, cell cycle arrest or apoptosis	Cancer (tumorsupressor) and Down syndrome (potential)	[15,72]
PAHX-AP1	Phytanoyl-CoA α-hydroxylase-associated protein 1	I, S	Facilitate DYRK1A-CREB interaction and development of neurological abnormalities	Down syndrome (potential)	[33]
Prkn	Parkin	I, S	E3 ubiquitin ligase activity and neuronal protection inhibition and loss of dopaminergic neurons	Parkinson’s disease	[73]
PS1	Presenilin1	I, S	Increase γ-secretase activity and accumulation of β- amyloid peptides (Aβ) in plaques	Down syndrome and Alzheimer’s disease	[74]
RCAN1	Regulator of calcineurin 1	I	Dysregulation of calcineurin, inhibition of signaling pathways that are controlled by NFAT and thus tau dysregulation, neuronal apoptosis, cell proliferation and development, etc.	Down syndrome, Alzheimer’s disease, and cancer	[75,76]
SRSF2/SC35	Splicing factor 35	I, S	Dysregulation of tau exon 10 splicing, imbalance of 3R-tau and 4R-tau expression and neurofibrillary degeneration	Down syndrome, Alzheimer’s, and Parkinson’s disease	[32]
SEPT4	Septin 4	I, S	Accumulation of β- amyloid peptides (Aβ) in plaques, tau self-aggregation and fibrillation, aggregation/inclusion formation of α-synuclein (Lewy bodies), loss of dopaminergic neurons.	Down syndrome (potential), Alzheimer’s, and Parkinson’s diseases (potential)	[77]
SIRT1	Sirtuin 1	I, S	Cell apoptosis inhibition	Down syndrome (potential), cancer (oncogene)	[19]
STAT3	/	I, S	Disruption of DNA damage, ROS regulation and cell death in leukemic B cells	Leukemia	[57,58]
τ/Tau	Tau protein	I, S	Reduction of tau biological activity and tau self-aggregation and fibrillation	Alzheimer’s disease, Parkinson’s disease, Fronto-temporal dementia	[78]
TOM70	Translocase of the outer mambrane	I	Decrease in import capacity of metabolite carriers and critical problem in mitochondrial machinery	Metabolic diseases, MRD7 and Down syndrome (potential)	[79]
TRAF3		I, S	Degradation of the noncanonical nuclear factor (NF)-κB–inducing kinase (NIK)	Autoimmune disease and leukemia (potential)	[80]
α-syn	α-synuclein	I, S	Aggregation/inclusion formation of α-synuclein (Lewy bodies) and loss of dopaminergic neurons	Parkinson’s disease	[81]

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
