# Peer review of "The Omnipresence of DYRK1A in Human Diseases"

_ijms, 2022, doi:10.3390/ijms23169355_

Round 1

Reviewer 1 Report

Deboever and colleagues have written a revision manuscript on the links between the protein kinase DYRK1A and disease in humans. Several recent reviews cover this issue (period 2020-2022): cancer, PMID: 35220406, 32870330, 32751160; therapeutic potential, PMID: 34954592; neurodevelopment and cognition, PMID: 34828439; neurodegeneration: PMID: 34445804; disease, PMID: 34205123; Down syndrome, PMID: 30268771. Therefore, the review is not particularly novel or covers unreview aspects of this protein. Moreover, I find many inaccuracies in the text, as the ones listed below. Finally, the authors should go carefully through some of their statements to avoid sending messages of proven/established experimental links or associations, when in some cases the published data is only suggestive of such links.

1.- Table 1 needs to be deeply amended:

- Table 1 is misleading because it can be interpreted as the list of proteins included as direct targets (substrates and/or interactors) of DYRK1A, which is not the case for several of them, including: c-MET, EGFR, HP1, MEK, NEP. Some clarification notes will help

- The other element of confusion with Table 1 is the direct link established with human diseases, which in several cases is just a suggestion, with no proofs of causal relationships. In these cases the disease link, particularly for Down syndrome (DS), is just simply suggested because the target protein is expressed in brain, and not even a role in DS has been proven for the target. The table therefore sends a dangerous message. These would be the case of: AMPH, Arip4, B-Raf, CREB1, Dcaf7, DREAM complex (for DS), GSK3, HP1, MEK, Notch1, NRSF, P27 (DS), p53 (DS), PAHX-AP1, SEPT4, SIRT1 (DS). Some clarification notes will help.

- no functional links between GRB2 and DYRK1A have been reported. The reference described alterations in protein amounts in a polytransgenic mouse cell line. It should not be included.

- missing links: EGFR and glioblastoma (PMID: 23635774), why not ID2 and cancer? (PMID: 26735018), why not FOXO and STAT3 and leukemia (PMID: 33393494), why not TOM70 (PMID: 34257281) and metabolic diseases?, why not TRAF3 (PMID: 34255829) and autoimmunity?

- suggestion of not including reviews in the list of references and maintained original papers: DREAM complex (#19, #33, #34, should be eliminated and add the original publication connecting DYRK1A with some components of the DREAM complex, PMID: 21498570) GSK3 (eliminate ref#38), HP1 (eliminate ref#42), NRSF (why not providing the original reference for the study instead of a review as for the other cases?), P27 (why not providing the original reference for the study instead of a review as for the other cases?), p53 (eliminate ref#19), RCAN1 (eliminate ref#53).

- I suggest to eliminate SR, because several of the SR proteins are individually listed in the table

- synuclein reference is wrong (PMID: 16959772)

- I suggest the authors to use official and current symbols or at least both: SRSF7/9G8, RAD54L2/Arip4, SRSF1/ASF, MME/NEP, CDKN1B/p27, SRSF2/SC35, t/Tau

2.- Subheading 1

- line 34, I suggest to add the original reference of the human kinome (PMID: 12471243) together with ref#6

- line 37, ref#6-8 are not really appropriate for the sentence on how kinases work in general. Ref#7 is repeated as ref#10

- line 35, symbol for the transferred phosphate is wrong

- line 39, I suggest to substitute ref#6 by PMID: 34354255, a recent overview of kinases as targets in disease

3.- Subheading 2

- line 47, the CMGC family was already defined in PMID: 12471243 not in ref#9

- line 48, it would be more correct to say: Within the CMGC family, the  human DYRK branch.

- line 51, members of the DYRK family are present from yeast to humans, therefore the sentence “insects-to-humans” is incorrect 

- Layout of Figure 1, shows “Nucleus” as the subcellular localization for DYRK1B, however cytosolic localization has been also shown for this protein in human cells (see for instance PMID: 16618736).

4.- Subheading 3

- It is named “DYRK1A expression and enzymatic activity…”, but no information on DYRK1A expression is provided, to whether is tissue specific or developmentally regulated.

- line 94, provide appropriate references as PMID: 11672423, 15960979

- PMID: 11672423 also for the effect of the substitution of Y321F

- line 97, I suggest to keep ref#62, which is the original reference.

- line 98, I suggest to add, “at least in vitro”, since the results in ref#62 were performed with a truncated protein expressed and purified from bacteria, and not proven to occur in vivo.

- DYRK1A is mainly localized in the cytosol in endogenous conditions, please check, for instance, subcellular fractionation experiments in PMID: 25620562, 19722700; CNS immunohistochemistry in PMID: 12576186, 12814361, among the many examples. It is accumulated in the nucleus when exogenously overexpressed.

- I do not understand the logic of assigning different “Review” references in lines 101, 103, 104, for each sentence. I do not understand the reasons for including references to original works, as #18, #64.

5.- Subheading 4

- DS and MRD7 are not neurodegenerative disorders, and they involve more than CNS alterations.

- line 113, MRD7 is also known as DYRK1A haploinsufficiency syndrome, and as such should be also mentioned.

- line 121, it is not correct to provide ref#67, centered on DYRK1A, as a reference for DS. The authors can choose one or two among the many good reviews available for the syndrome.

- line 121, MRD7 is a much more complex syndrome that the description provided; references should be included (for instance, PMID: 25707398, 34345024). Should it be better to introduce MRD7 after the mapping of DYRK1A within chromosome 21 and its link to DS?

- line 123, not clear what the authors mean with “both phenotypes are affected in similar brain areas”,  which phenotypes???

- the authors should not forget that MRD is a monogenic disease, and DS is a polygenic disease, and that not all the DS clinical phenotypes can be assigned to dysregulation of DYRK1A; in fact, very few have been experimentally proven by making Dyrk1a disomic in a trisomic context and leading to the correction of the phenotype under study.

- line 132, dosage sensitivity in SNC development has been also shown in retinal development and associated-defects (PMID: 19081073).

- line 146, not clear to “which relationship” the authors are referring in MRD7 patients

- the authors are missing the fact that Dyrk1a heterozygous mouse models recapitulate several of the neurodevelopmental traits in MRD7 patients as shown in PMID: 30831192, 29223763.

- the authors focus the DYRK1A-dependent effects on the impact on neuronal progenitors while not mentioning DYRK1A role on  post-mitotic neuronal activities, such as dendritogenesis or synaptic function, which are critical for brain homeostasis (see for instance 35194165 among many examples)

- line 149, it should say “microcephaly” not “microencephaly”

6.- Subheading 4.2

- line 162, ref for Kimura et al is missing

- line 168, ref should be PMID:18509201

- I find too extensive the description of the alterations in tau splicing, while important work linking DYRK1A and AD in animal models is not discussed at all (for instance 28647555, 28779511, 31267651, 33380426).

- not clear which is the link between RCAN1, NFAT, DYRK and dementia and AD.

- very little information on the effects of DYRK1A on APP (35803734, 35835549)

7.- Subheading 4.3

- line 221, ref#78 does not show any result on DYRK1A expression in PD, HD and FTD.

- line 221, I would not call a work from 2005 (ref#79) a recent work. This reference shows OE of DYRK1A in AD, but not in PD/HD or FTD

- missing info on in vivo models of PD as 24922073, 31572127

8.- Subheading 5.1

- I suggest to provide general references for T1D of T2D as it has been the case for the other diseases in previous sections. Ref#81 does not seem appropriate

- please, check this very recent publication PMID:35700053, to include relevant information for this section.

9.- Subheading 5.2

- line 274, I do not understand the reason for stressing the epigenetic face of cancer in the context of DYRK1A, since there are not reported reasons for such a link.

- line 276, the authors forget to mention that while DS individuals have increased risk of developing certain type of cancers, particularly leukemia, they showed reduced risk for many other types of solid tumors . The authors’ sentence sends therefore a wrong message to the reader. Same wrong message is in the next sentence implying that DYRK1A is a tumor promoter in DS individuals. Current information suggests that the kinase likely has a dual role as both tumor suppressor and tumor promoter in the context of DS (22354171, 19458618), as the authors discussed in the next paragraph.

10.- Subheading 5.3

- No mention of DYRK1A association with the biology of oncogenic viruses as adenovirus and HPV?

11.- Subheading 5.4

- line 335, ref is missing. I suppose the authors refer to 35810562

12.- Subheading 6

- Figure 3, I think that there is no enough experimental information to establish a link between DYRK1A and Huntington’s disease or fronto-temporal dementia. Why the authors distinguish between cancer and leukemia? Leukemia is a type of cancer.

- line 392, missing 35053488, 31024071

Author Response

We thank all the reviewers for the time and hard work dedicated reviewing our article. We have endeavored to take each comment and suggested improvement into account in the accompanying revision. Below we summarize changes made in response to each comment. Please see all the adaptations made in the revised manuscript.

Reviewer 1

  1. Deboever and colleagues have written a revision manuscript on the links between the protein kinase DYRK1A and disease in humans. Several recent reviews cover this issue (period 2020-2022): cancer, PMID: 35220406, 32870330, 32751160; therapeutic potential, PMID: 34954592; neurodevelopment and cognition, PMID: 34828439; neurodegeneration: PMID: 34445804; disease, PMID: 34205123; Down syndrome, PMID: 30268771. Therefore, the review is not particularly novel or covers unreview aspects of this protein. Moreover, I find many inaccuracies in the text, as the ones listed below. Finally, the authors should go carefully through some of their statements to avoid sending messages of proven/established experimental links or associations, when in some cases the published data is only suggestive of such links.

We thank reviewers #1 and #2 for this understandable comment. Indeed, two review papers have been previously published in Int. J. Mol. Sci. with related subjects. The first one (https://doi.org/10.3390/ijms22169083) focuses on the impact of DYRK1A kinase on diabetes while the second review (https://doi.org/10.3390/ijms22116047) gives an overview of the roles of both the DYRK and CLK family of kinases in human diseases. These two review articles are included in the references of this review as the ones mentioned by reviewer #1.

As mentioned by reviewer 1, other reviews exist but these focus either on all members of the DYRKs family or on a single disease. The originality of our review is that it focuses only on one member of the family, DYRK1A, plus is interested in all the human pathologies affected by this protein (expression, activity, etc.). By focusing only on the DYRK1A kinase, our review adds to the recent literature because it clearly highlights the ubiquity of this protein in the development of various human diseases, not only diabetes or neurodegenerative diseases as is regularly observed in the literature. Our vision is innovative and extensive as it allows the reader to have a wide view of the impact of this protein but also to understand more about the biochemistry of DYRK1A, if known, behind each pathology. In contrast to the other two reviews mentioned by reviewer #2, which are more generalized, our review is much more detailed on the mechanisms of action and the pathologies presented. The goal of our review is to provide the reader with a quick but detailed overview of the situation without the need to go through numerous articles. Of course, all the references provided by our review allow the reader to dig deeper if needed.

Finally, as scientific research on this kinase family has advanced rapidly in recent years, an update was necessary to inform the reader about new applications, especially in the development of viral diseases.

  1. Table 1 needs to be deeply amended: Table 1 is misleading because it can be interpreted as the list of proteins included as direct targets (substrates and/or interactors) of DYRK1A, which is not the case for several of them, including: c-MET, EGFR, HP1, MEK, NEP. Some clarification notes will help

We thank reviewer #1 for noticing this possible misinterpretation of Table 1. We have taken this comment into account by giving information on the function (substrate and/or interactor) and the detailed effects of each DYRK1A target. Please see the modification directly in the revised manuscript as many modifications have been made in the revised Table 1.

  1. The other element of confusion with Table 1 is the direct link established with human diseases, which in several cases is just a suggestion, with no proofs of causal relationships. In these cases the disease link, particularly for Down syndrome (DS), is just simply suggested because the target protein is expressed in brain, and not even a role in DS has been proven for the target. The table therefore sends a dangerous message. These would be the case of: AMPH, Arip4, B-Raf, CREB1, Dcaf7, DREAM complex (for DS), GSK3, HP1, MEK, Notch1, NRSF, P27 (DS), p53 (DS), PAHX-AP1, SEPT4, SIRT1 (DS). Some clarification notes will help.

Based on the reviewer’s comment, we have adapted Table 1 by adding some more information on the subject. If the link with the disease is only suggested and/or not already proved, the indication “potential” was added next to the disease. Please see the modification directly in the revised manuscript as many modifications have been made in the revised Table 1.

  1. no functional links between GRB2 and DYRK1A have been reported. The reference described alterations in protein amounts in a polytransgenic mouse cell line. It should not be included.

After a thorough analysis of the above-mentioned article, it seems that the link has indeed not been shown but only suggested. We have modified Table 1 in accordance with this remark. Please see the modification directly in the revised manuscript as many modifications have been made in the revised Table 1.

  1. missing links: EGFR and glioblastoma (PMID: 23635774), why not ID2 and cancer? (PMID: 26735018), why not FOXO and STAT3 and leukemia (PMID: 33393494), why not TOM70 (PMID: 34257281) and metabolic diseases?, why not TRAF3 (PMID: 34255829) and autoimmunity?

We thank reviewer #1 for noticing these significant lacks in the references of Table 1. We have modified the reference in Table 1 in accordance with this remark. Please see the modification directly in the revised manuscript as many modifications have been made in the revised Table 1.

  1. suggestion of not including reviews in the list of references and maintained original papers: DREAM complex (#19, #33, #34, should be eliminated and add the original publication connecting DYRK1A with some components of the DREAM complex, PMID: 21498570) GSK3 (eliminate ref#38), HP1 (eliminate ref#42), NRSF (why not providing the original reference for the study instead of a review as for the other cases?), P27 (why not providing the original reference for the study instead of a review as for the other cases?), p53 (eliminate ref#19), RCAN1 (eliminate ref#53).

We have modified the references in Table 1. Please see the modification directly in the revised manuscript as many modifications have been made in the revised Table 1

  1. I suggest to eliminate SR, because several of the SR proteins are individually listed in the table

According to this remark, we removed SR from Table 1.

  1. synuclein reference is wrong (PMID: 16959772)

We have corrected the reference in Table 1.

  1. I suggest the authors to use official and current symbols or at least both: SRSF7/9G8, RAD54L2/Arip4, SRSF1/ASF, MME/NEP, CDKN1B/p27, SRSF2/SC35, t/Tau

According to this remark, symbols have been modified in Table 1.

  1. Subheading 1, line 34, I suggest to add the original reference of the human kinome (PMID: 12471243) together with ref#6

The reference was added according to reviewer #1’s suggestion. Please see the adaptation directly in the revised manuscript as all the references of the article have been updated.

  1. line 37, ref#6-8 are not really appropriate for the sentence on how kinases work in general.

The references were adapted according to reviewer #1’s suggestion. Please see the adaptation directly in the revised manuscript as all the references of the article have been updated.

  1. Ref#7 is repeated as ref#10

This duplicate reference has been corrected.

  1. line 35, symbol for the transferred phosphate is wrong

We thank reviewer #1 for noticing this error, the sentence was corrected according to that. Please see the modification as follow in the revised manuscript (p.1 lines 34-37):

“Protein kinases (PK) are responsible for protein phosphorylation by catalyzing the transfer of a-phosphate of ATP to the hydroxyl groups (on serine and threonine for serine/threonine kinases) or phenolic groups (on tyrosine for tyrosine kinases) of proteins to generate phosphate monoesters [5,7].”

  1. line 39, I suggest to substitute ref#6 by PMID: 34354255, a recent overview of kinases as targets in disease

This reference has been changed per the reviewer’s suggestion.

  1. Subheading 2, line 47, the CMGC family was already defined in PMID: 12471243 not in ref#9

The reference was adapted according to reviewer #1’s suggestion. Please see the adaptation directly in the revised manuscript as all the references of the article have been updated.

  1. line 48, it would be more correct to say: Within the CMGC family, the humanDYRK branch.

The sentence has been adapted according to reviewer #1’s suggestion. Please see the modification in the revised manuscript as follow (p.2 lines 50-52):

“Within the CMGC family, the human DYRK (dual-specificity tyrosine-regulated kinase) branch comprises 5 members divided into 2 classes.”

  1. line 51, members of the DYRK family are present from yeast to humans, therefore the sentence “insects-to-humans” is incorrect 

We thank reviewer #1 and #3 for noticing that this sentence could be misunderstood. We have adapted the sentence based on these suggestions. Please see the modification as follow (p.2 lines 53-55):

“From yeast to humans, and recently also discovered in fish, DYRKs are highly conserved, as orthologous genes have been cloned independently in various eukaryotic organisms (Figure 1) [10,12,13]

  1. Layout of Figure 1, shows “Nucleus” as the subcellular localization for DYRK1B, however cytosolic localization has been also shown for this protein in human cells (see for instance PMID: 16618736).

The PDB database used for this figure failed to mention this. We have carefully adapted the figure 1 according to this suggested reference and the reference was added to the revised manuscript.

  1. Subheading 3. It is named “DYRK1A expression and enzymatic activity…”, but no information on DYRK1A expression is provided, to whether is tissue specific or developmentally regulated.

We thank reviewers #1 for this understandable comment also suggested by reviewer #3. We have carefully taken all these comments into consideration, and we have adapted this section based on these suggestions. Please see the modifications made in the revised manuscript as follow (p. 8 lines 91-123):

“From kidneys to bone marrow, DYRK1A expression is ubiquitously observed in all human tissues from early embryonic development to adulthood [72,73]. High expression of Dyrk1A was detected in several areas of the adult hindbrain, particularly in the cerebellum and functionally related structures such as the precerebellar and cerebellar nuclei and the vestibular nuclear complex [74]. Subcellular localization of the kinase, however, is still the focus of numerous studies as, in addition to a PEST sequence at its C-terminus (Figure 2), Dyrk1A has a nuclear targeting sequence at its N-terminus. However, studies showed that a substantial amount of endogenous Dyrk1A is localized within the cytoplasm of the cells in the brains of humans, mice, and chickens [75,76]. This dual localization is due to the fact that DYRK1A substrates are both nuclear and cytosolic proteins [10,76] While DYRK1A is mainly localized in the cytosol in endogenous conditions, it is also accumulated in the nucleus when exogenously overexpressed from substrates that shuttle to and from the nucleus [53,58,66,76].

Like most other kinases, DYRK1A depends on a molecular switch to adopt an active/inactive conformation. Whereas the dual-phosphorylation of the MAPKs is a classical paradigm for the on⁄off regulation by upstream protein kinases, DYRKs rely on auto-phosphorylation of an absolutely conserved tyrosine (Y321 for DYRK1A) residue in the activation loop (see Figure 2) and appears to be an evolutionarily conserved ancestral feature 61. It was shown that, for mammalian DYRK1A, tyrosine autophosphorylation is an intrinsic capacity of the catalytic domain and does not depend on other domains or any cofactor. However, Y321 substitution with phenylalanine significantly reduces, at least in vitro, the catalytic activity of DYRK1A and, surprisingly, dephosphorylation does not inactivate mature DYRK1A [22,77]. It is therefore assumed that tyrosine phosphorylation is only required for activation but not for maintenance of the active state.

Based on these findings, DYRK1A has multiple biological functions: signaling, mRNA splicing, chromatin transcription, DNA damage repair, cell survival, cell cycle control, neuronal development and functions, synaptic plasticity, etc. [10–12,19,78]. The level of activity of DYRK1A is of key importance for its physiological and pathological effects [24,79,80]. Hence, DYRK1A is regulated based on both gene expression and protein abundance (for reviews see [77–79,81]) with both low and high expression exerting significant effects on human diseases (Table 1). Due to its low tissue specificity, multiple proteins are targeted by DYRK1A overexpression, leading to a multitude of various changes within the cell and, consequently, to a plethora of different symptoms and pathologies”.

  1. line 94, provide appropriate references as PMID: 11672423, 15960979. PMID: 11672423 also for the effect of the substitution of Y321F

The suggested references have been added.

  1. line 97, I suggest to keep ref#62, which is the original reference.

The reference remains in the revision.

  1. line 98, I suggest to add, “at least in vitro”, since the results in ref#62 were performed with a truncated protein expressed and purified from bacteria, and not proven to occur in vivo.

The sentence has been adapted according to reviewer #1’s suggestion. Please see the modification in the revised manuscript as follow (p.8 lines 111-113):

“However, Y321 substitution with phenylalanine significantly reduces, at least in vitro, the catalytic activity of DYRK1A and, surprisingly, dephosphorylation does not inactivate mature DYRK1A [22,77].”

  1. DYRK1A is mainly localized in the cytosol in endogenous conditions, please check, for instance, subcellular fractionation experiments in PMID: 25620562, 19722700; CNS immunohistochemistry in PMID: 12576186, 12814361, among the many examples. It is accumulated in the nucleus when exogenously overexpressed.

We thank reviewers #1 for this understandable comment also suggested by reviewer #3. We have carefully taken all these comments into consideration, and we have adapted this section based on these suggestions. Please see the modifications made in the revised manuscript as already presented in response to remark #19.

  1. I do not understand the logic of assigning different “Review” references in lines 101, 103, 104, for each sentence. I do not understand the reasons for including references to original works, as #18, #64.

The references were adapted according to reviewer #1’s suggestion. Please see the adaptation directly in the revised manuscript as all the references of the article have been updated.

  1. Subheading 4. DS and MRD7 are not neurodegenerative disorders, and they involve more than CNS alterations.

We thank reviewer #1 for noticing that this subheading could be misunderstood. We have adapted the subheading according to this remark. Please see the modification in the revised manuscript as follow (p.8 line 124):

“4. DYRK1A and neurological diseases”.

  1. line 113, MRD7 is also known as DYRK1A haploinsufficiency syndrome, and as such should be also mentioned.

The sentence has been adapted based on the reviewer’s comment. Please see the adaptation made in the revised manuscript as follow (p.9 lines 141-143):

“In the same way, patients with DYRK1A haploinsufficiency syndrome, also known as MRD7, experience a heterozygous loss-of-function in DYRK1A.”

  1. line 121, it is not correct to provide ref#67, centered on DYRK1A, as a reference for DS. The authors can choose one or two among the many good reviews available for the syndrome.

The references were adapted according to reviewer #1’s suggestion. Please see the adaptation directly in the revised manuscript as all the references of the article have been updated.

  1. line 121, MRD7 is a much more complex syndrome that the description provided; references should be included (for instance, PMID: 25707398, 34345024). Should it be better to introduce MRD7 after the mapping of DYRK1A within chromosome 21 and its link to DS?

Based on reviewer #1 suggestion, we have carefully adapted this section in the manuscript to offer a better description of MRD7 syndrome and we added the suggested articles into the references of our review. We have also reorganized the section according to the reviewer’s remark and we have introduced MRD7 after mapping DYRK1A gene within chromosomes and explaining DS. Please see the modifications made in the revised manuscript as follow (p. 8-9 lines 133-155):

“With an incidence of 1 in 800 lives births, DS, which is caused by a trisomy of chromosome 21, is one of the leading causes of intellectual disability. In addition to all the problems of cognitive development, patients with DS face various health issues throughout their lives, including learning and memory deficits, congenital heart disease, AD, leukemia, and cancer, leading to huge medical and social costs [83].

The DYRK1A gene is located within human chromosome 21q22.2, also known as the Down Syndrome Critical Region (DSCR). There is considerable genetic and pharmacologic evidence showing that the mere 1.5-fold overexpression of DYRK1A is responsible for a variety of symptoms observed in DS patients. In the same way, patients with DYRK1A haploinsufficiency syndrome, also known as MRD7, experience a heterozygous loss-of-function in DYRK1A. This syndrome is characterized by severe intellectual dis-ability, speech and motor delay, autism spectrum disorder and sometimes epileptic seizures (for complete reviews see [84–86]). Both DS and MRD7 syndromes affect in similar brain areas and functions; but MRD7 is distinguished by the observation of pronounced microcephalies [82,87]. MRD7 is a monogenic disease whereas DS is a polygenic disease and not all the DS clinical phenotypes are assigned to dysregulation of DYRK1A. However, these diseases demonstrate the critical dosage sensitivity of DYRK1A as either too much or too little kinase activity has negative health consequences [83]. This is likely due to the critical role of DYRK1A in neural development and timing of neural progenitor cell differentiation. Indeed, Najas and co-workers showed, in the Ts65Dn mouse model, that a 1.5-fold increase in DYRK1A protein levels in neural stem cells of the developing cerebral cortex caused by trisomy of the DYRK1A gene lengthens the cell cycle and decreases the production of neurons [88].”

  1. line 123, not clear what the authors mean with “both phenotypes are affected in similar brain areas”,  which phenotypes???

We have adapted the sentence according to this remark. Please see the modification in the revised manuscript as follow (p.9 lines 145-147):

“Both DS and MRD7 syndromes affect similar brain areas and functions; but MRD7 is distinguished by the observation of pronounced microencephalies [82,87].”

  1. the authors should not forget that MRD is a monogenic disease, and DS is a polygenic disease, and that not all the DS clinical phenotypes can be assigned to dysregulation of DYRK1A; in fact, very few have been experimentally proven by making Dyrk1a disomic in a trisomic context and leading to the correction of the phenotype under study.

Based on reviewer #1 previous suggestions, we have carefully adapted this section in the manuscript to offer a better description of MRD7 syndrome and we added the suggested articles into the references of our review. We have also included this remark into the revised manuscript. Please see the modifications made in the revised manuscript as previously presented in remark #28.

  1. line 132, dosage sensitivity in SNC development has been also shown in retinal development and associated-defects (PMID: 19081073).

We thank reviewer #1 for this suggestion and we have adapted our paragraph according to this. Please see the modifications made in the revised manuscript as follow (p. 9 lines 163-170):

“The deficit in neuron numbers in the trisomic Ts65Dn mouse is restored by normalizing DYRK1A gene-dosage [86]. Additionally, the neural retina is thicker in DS individuals than in the normal population. Increased retinal size and cellularity in Ts65Dn mouse correlated with abnormal retinal function and resulted from an impaired caspase-9-mediated apoptosis during development. It was shown that normalization of DYRK1A gene copy number in Ts65Dn mice rescues both morphological and functional phenotypes [33,34]. These observations suggest that DYRK1A inhibition could have some therapeutic value for DS patients.”

  1. line 146, not clear to “which relationship” the authors are referring in MRD7 patients

By “relationship”, we wanted to express the fact that DYRK1A dosage was crucial for the development of DS, but unfortunately no parallel had yet been demonstrated for the case of MRD7 syndrome. Based on this and in order to make our comments clearer, we have adapted the sentence. Please see the modifications made in the revised manuscript as follow (p. 9 lines 170-176):

“To date, no such parallel has been identified in MRD7 patients, although it has been shown in heterozygous DYRK1A +/- mice that such a reduction in protein expression was systematically associated with significant reduction in body weight and length and pronounced microcephaly, thus mirroring some neurological traits associated with the human MRD7 pathology, such as defective social interactions, stereotypic behaviors and epileptic activity, with also altered proportions of excitatory and inhibitory neocortical neurons and synapses [84,88,89].”

  1. the authors are missing the fact that Dyrk1a heterozygous mouse models recapitulate several of the neurodevelopmental traits in MRD7 patients as shown in PMID: 30831192, 29223763.”

According to this remark, we have adapted our paragraph to add more information about this mouse model and we have included the two suggested articles into our references. Please see the adaptations made in the revised manuscript as presented in our previous response (#31).

  1. the authors focus the DYRK1A-dependent effects on the impact on neuronal progenitors while not mentioning DYRK1A role on  post-mitotic neuronal activities, such as dendritogenesis or synaptic function, which are critical for brain homeostasis (see for instance 35194165 among many examples)

We are not unaware of these effects on functioning of mature neurons. However, the focus on progenitors is done to emphasize the critical role of Dyrk1a copy number and the resulting neurodevelopmental diseases that arise when there is either too little DYRK1A or too much DYRK1A.

  1. line 149, it should say “microcephaly” not “microencephaly”

This issue has been corrected.

  1. Subheading 4.2, line 162, ref for Kimura et al is missing

This reference has been added.

  1. line 168, ref should be PMID:18509201

The reference was adapted according to reviewer #1’s suggestion. Please see the adaptation directly in the revised manuscript as all the references of the article have been updated.

  1. I find too extensive the description of the alterations in tau splicing, while important work linking DYRK1A and AD in animal models is not discussed at all (for instance 28647555, 28779511, 31267651, 33380426).

According to the reviewer’s comment, we have adapted this section of the manuscript by adding more information on the effects of DYRK1A through mouse model studies. Please see the adaptation made in the revised manuscript as follow (p. 9-10 lines 185-207):

“From this perspective, recent evidence implicates DYRK1A as a promising target for AD as it has been associated with both tau and amyloid neuropathologies through DYRK1A-mediated phosphorylation of key substrates: tau, amyloid precursor protein (APP), Neprilysin (NEP), and Presenilin 1 (PSEN1) and SEPT4 (Table 1). A recent work highlights DYRK1A as a modulator of the axonal transport machinery driving APP intracellular distribution in neurons. Indeed, DYRK1A inhibition affects the expression (transcription, translation and/or accumulation) of proteins that are involved in the intracellular trafficking of APP loaded vesicles [93]. Consistent with this, inhibition of the kinase in AD transgenic mouse models decreased APP and amyloid-b accumulation [94–96]. However, in the Dp3Tyb 528 model, which contains an additional copy of DYRK1A, no increase in APP abundance or sign of amyloid-b accumulation was observed [97], suggesting that having 3 copies of DYRK1A gene is not sufficient to modulate APP protein abundance or promote amyloid-β accumulation.

In 2007, Kimura et al. showed in HEK293 cells that the amount of tau phosphorylated at Thr212 increased by co-transfection of the DYRK1A expression vector, whereas tau level was similar [68]. Tau phosphorylation at Thr212 is associated with Aβ overproduction. Moreover, it was shown that this tau phosphorylation at Thr212 primes tau for phosphorylation by glycogen synthase kinase-3 beta (GSK3-β) at Ser208, contributing to the formation of paired helical filaments composed of highly phosphorylated tau, a component of neurofibrillary tangles (NFT) [68,98]. In this context, in vivo studies were conducted in various mouse models and all compounds tested were able to reduce tau phosphorylation through inhibition of DYRK1A activity, leading to inhibition of tau oligomerization and aggregation [99,100].”

  1. not clear which is the link between RCAN1, NFAT, DYRK and dementia and AD.

We thank reviewer #1 for this remark. We have adapted this paragraph to make it clearer for the reader. Please see the adaptation made in the revised manuscript as follow (p. 10 lines 226-239):

“On the other hand, tau dephosphorylation has been observed resulting from increases of both DYRK1A and regulator of calcineurin 1 (RCAN1) expression in human DS and AD brains. Calcineurin is a calcium/calmodulin dependent serine/threonine phosphatase which promotes, by dephosphorylating the nuclear factor of activated T cells (NFAT) transcription factor, NFAT translocation into the nucleus contributing to several genes’ transcription and subsequent events (e.g., cell proliferation, apoptosis, angio-genesis, synaptic plasticity, immune response, and skeletal/cardiac muscle development) [64]. However, RCAN1 overexpression results from the inhibition of signaling pathways that are controlled by NFAT (Table 1) which was shown to directly interact with DYRK1A. Recently, DYRK1A was demonstrated to directly interact with and phosphorylate RCAN1 at Ser112 and Thr192 residues, which would prime the protein for further phosphorylation by other protein targets like GSK3-b at Ser108 [65]. This synergistic interaction of DYRK1A, NFAT and RCAN1 could contribute to a variety of pathological features of DS, including early onset of AD [64,65].”

  1. very little information on the effects of DYRK1A on APP (35803734, 35835549)

We have adapted this section of the manuscript by adding more information on the effects of DYRK1A on APP, notably through mouse model studies. Please see the revised manuscript as presented in our previous response for remark #37.

  1. Subheading 4.3, line 221, ref#78 does not show any result on DYRK1A expression in PD, HD and FTD. I would not call a work from 2005 (ref#79) a recent work. This reference shows OE of DYRK1A in AD, but not in PD/HD or FTD

The reference was adapted according to reviewer #1’s suggestion. Please see the adaptation directly in the revised manuscript as all the references of the article have been updated.

  1. missing info on in vivo models of PD as 24922073, 31572127

According to the reviewer’s comment, we have adapted this section of the manuscript by adding more information on the effects of DYRK1A on PD, notably through mouse model studies. Please see the adaptation made in the revised manuscript as follow (p. 11 lines 262-278):

“Recently, it has been demonstrated that increased DYRK1A levels were present in the brains of patients not only with AD but as well as other neurodegenerative diseases such as PD, HD and FTD syndromes [106,107]. Indeed, DYRK1A-mediated phosphor-ylation of Parkin (Table 1) inhibits its E3 ubiquitin ligase activity and, consequently, impairs its neuroprotective function in dopaminergic neurons [62]. Likewise, SEPT4, which has been found in neurofibrillary tangles and in a-synuclein-positive cytoplasmic inclusions (Lewy bodies) in PD brains, is phosphorylated by DYRK1A (Table 1) which leads to more α-synuclein (α-syn) aggregation and loss of dopaminergic neurons [67]. This phenomenon is accentuated by the fact that DYRK1A can also directly phosphorylate α-syn, contributing to its aggregation and, potentially, to disease progression [71]. This time, in vivo assays demonstrated that Dyrk1a expression is crucial for the survival of dopaminergic neurons (DA) in the MPTP-induced mouse model of PD due to an abnormal activity of the mitochondrial caspase9 (Casp9)-dependent apoptotic pathway during the main wave of programmed cell death that affects these neurons [108]. Another study, performed on the same PD mouse model, suggested that an increased level of miR-204 results in the death of DA by upregulating the expression of DYRK1A and targeting the DYRK1A-mediated apoptotic signaling pathway [109].].”

  1. Subheading 5.1, I suggest to provide general references for T1D of T2D as it has been the case for the other diseases in previous sections. Ref#81 does not seem appropriate

The reference was adapted according to reviewer #1’s suggestion. Please see the adaptation directly in the revised manuscript as all the references of the article have been updated.

  1. please, check this very recent publication PMID:35700053, to include relevant information for this section.

This section has been adapted to include this new relevant information. Please see the modification made in the revised manuscript as follow (p. 12 lines 319-324):

“A recent study revealed the importance of the DREAM complex in enforcing quiescence in adult human β cells and demonstrated that small molecule DYRK1A inhibitors induce human β cells to replicate by converting the repressive DREAM complex into an alternate pro-proliferative configuration containing the MuvB complex and B-MYB (MYBL2) referred to as the “MMB complex” [110]. Although encouraging, the community still awaits clinical entry of a DYRK1A inhibitor for T1D.”

  1. Subheading 5.2. line 274, I do not understand the reason for stressing the epigenetic face of cancer in the context of DYRK1A, since there are not reported reasons for such a link.

We thank reviewer #1 for noticing that this paragraph was unclear on the subject. We have adapted the paragraph based on that and to better match our statement. Please see the modifications made in the revised manuscript as follow (p. 12 lines 326-332):

“Development of malignant cells through aberrant gene function and altered gene expression patterns has been the focus of medical research in recent years while dissecting the origins of cancers and leukemias. Growing evidence shows that epigenetic factors are primarily involved in causing these abnormalities, along with genetic alterations [119,120]. Considering the increasing number of emerging tumors (lymphomas, retinoblastomas, etc.) in patients with DS, researchers have investigated the possible link between this genetic disorder and the development of these malignant cells [50].”

  1. line 276, the authors forget to mention that while DS individuals have increased risk of developing certain type of cancers, particularly leukemia, they showed reduced risk for many other types of solid tumors . The authors’ sentence sends therefore a wrong message to the reader. Same wrong message is in the next sentence implying that DYRK1A is a tumor promoter in DS individuals. Current information suggests that the kinase likely has a dual role as both tumor suppressor and tumor promoter in the context of DS (22354171, 19458618), as the authors discussed in the next paragraph.

We have adapted the paragraph based on that and to better match our statement. Please see the modifications made in the revised manuscript as follow (p. 12 lines 336-338):

“However, despite numerous research reports in recent years, the role of DYRK1A in cancer and leukemia, in the context of DS, is still very unclear as both oncogenic and tumor suppressive roles have been reported [24].”

And also in p.13 lines 357-363:

“Taken together, these studies show that DYRK1A is an important kinase in multiple cancers [121]. However, its specific functional role appears to be cell-type and context-dependent. While DS individuals have increased risk of developing certain type of cancers, particularly leukemia, they showed reduced risk for many other types of solid tumors [122,123]. Appreciating these nuances of DYRK1A activity will be critical for the future pursuit of DYRK1A focused cancer therapeutics.”

  1. Subheading 5.3. No mention of DYRK1A association with the biology of oncogenic viruses as adenovirus and HPV?

We have adapted the section in order to add this important information. Please see the modification made in the revised manuscript as follow (p.13 lines 372-381):

“In the context of HIV-1t, DYRK1A regulates the activity of several transcription factors like NFAT and cyclin L2 (Table 1) thus making viruses latent and preventing replication in macrophages [55,124,125]. Moreover, DYRK1A interacts also with oncoproteins from adenovirus and human papillomavirus (HPV) [38,126]. For example, DYRK1A was shown to phosphorylates in vitro and in vivo Thr5 and Thr7 of HPV16E7, which is a high-risk tumorigenic viral protein identified as one of the causative agents for the development of cervical cancer [126]. Finally, a role of DYRK1A gene in the resistance against the severe acute respiratory syndrome coronavirus 2 (SARS-CoV-2), the causative agent of coronavirus disease 2019 (COVID-19), has been recently suggested [127]”

  1. Subheading 5.4, line 335, ref is missing. I suppose the authors refer to 35810562

We thank reviewer #1 for noticing that the supporting reference for that statement was missing. Please see the adaptation directly in the revised manuscript as all the references of the article have been updated.

  1. Subheading 6. Figure 3, I think that there is no enough experimental information to establish a link between DYRK1A and Huntington’s disease or fronto-temporal dementia. Why the authors distinguish between cancer and leukemia? Leukemia is a type of cancer.

We thank reviewer #1 for this understandable comment. Indeed, information on the impact of DYRK1A in these diseases are only suggestive for now. This is one of the reasons why our figure is a schematic representation of the DYRK1A spectrum and not a real illustration. Nevertheless, this information being unclear, we have adapted the legend of Figure 3 to correctly inform the reader about the message conveyed by this figure. Please see the modification made on Figure 3 directly in the revised manuscript.

Moreover, we do not see any distinction between cancer and leukemia in Figure 3. Only a distinction between the different possible cancers and leukemias is made; but they all start from the same section "cancer and leukemia".

  1. line 392, missing 35053488, 31024071

We thank reviewer #1 for noticing that these references for that statement were missing. Please see the adaptation directly in the revised manuscript as all the references of the article have been updated.

Reviewer 2 Report

- In 2021, two review paper with the similar topic has already been
published in Int. J. Mol. Sci.: 2021, 22(16), 9083; https://doi.org/10.3390/ijms22169083 and 2021, 22(11), 6047; https://doi.org/10.3390/ijms22116047. The authors have to explain why there is room for this new review and highlight what is the difference between this new review and previous ones?

- In addition, the authors should cite and comment on other previous reviews that have a similar or related topic, such as:
1. Oncogene (2022) 41:2003–2011; https://doi.org/10.1038/s41388-022-02245-6,
2. Int. J. Mol. Sci. 2021, 22(16), 9083; https://doi.org/10.3390/ijms22169083,
3. https://doi.org/10.1038/s41467-021-24426-9
4. Cells 2021, 10(9), 2263; https://doi.org/10.3390/cells10092263

-Abbreviations are not appropriately defined throughout the manuscript. I recommend preparing the list with abbreviations in the revised version of the ms

-The quality of the figures is quite low. Authors should include the high-resolution pictures in the revised version of ms.

 -The Authors should run spell check and carefully check for typos.

Author Response

We thank all the reviewers for the time and hard work dedicated reviewing our article. We have endeavored to take each comment and suggested improvement into account in the accompanying revision. Below we summarize changes made in response to each comment. Please see all the adaptations made in the revised manuscript.

Reviewer 2

  1. English language and style

( ) Extensive editing of English language and style required

( ) Moderate English changes required

(x) English language and style are fine/minor spell check required

( ) I don't feel qualified to judge about the English language and style

According to the reviewer’s comment, the English language was double checked, and minor spelling issues were carefully checked throughout the manuscript. Please see adaptations directly in the revised manuscript.

  1. In 2021, two review paper with the similar topic has already been published in Int. J. Mol. Sci.: 2021, 22(16), 9083; https://doi.org/10.3390/ijms22169083 and 2021, 22(11), 6047; https://doi.org/10.3390/ijms22116047. The authors have to explain why there is room for this new review and highlight what is the difference between this new review and previous ones?

We thank reviewers #1 and #2 for this understandable comment. Indeed, two review papers have been previously published in Int. J. Mol. Sci. with related subjects. The first one (https://doi.org/10.3390/ijms22169083) focuses on the impact of DYRK1A kinase on diabetes while the second review (https://doi.org/10.3390/ijms22116047) gives an overview of the roles of both the DYRK and CLK family of kinases in human diseases. These two review articles are included in the references of this review as the ones mentioned by reviewer #1.

As mentioned by reviewer 1, other reviews exist but these focus either on all members of the DYRKs family or on a single disease. The originality of our review is that it focuses only on one member of the family, DYRK1A, plus is interested in all the human pathologies affected by this protein (expression, activity, etc.). By focusing only on the DYRK1A kinase, our review adds to the recent literature because it clearly highlights the ubiquity of this protein in the development of various human diseases, not only diabetes or neurodegenerative diseases as is regularly observed in the literature. Our vision is innovative and extensive as it allows the reader to have a wide view of the impact of this protein but also to understand more about the biochemistry of DYRK1A, if known, behind each pathology. In contrast to the other two reviews mentioned by reviewer #2, which are more generalized, our review is much more detailed on the mechanisms of action and the pathologies presented. The goal of our review is to provide the reader with a quick but detailed overview of the situation without the need to go through numerous articles. Of course, all the references provided by our review allow the reader to dig deeper if needed.

Finally, as scientific research on this kinase family has advanced rapidly in recent years, an update was necessary to inform the reader about new applications, especially in the development of viral diseases.

  1. In addition, the authors should cite and comment on other previous reviews that have a similar or related topic, such as:
    • Oncogene (2022) 41:2003–2011; https://doi.org/10.1038/s41388-022-02245-6,
    • Int. J. Mol. Sci. 2021, 22(16), 9083; https://doi.org/10.3390/ijms22169083,
    • https://doi.org/10.1038/s41467-021-24426-9
    • Cells 2021, 10(9), 2263; https://doi.org/10.3390/cells10092263

We thank reviewer #2 for this suggestion and the relevant articles missing in the first version of our article. Based on the list of articles provided, we have carefully checked our references and added the missing ones into the revised manuscript. Please see the adaptation directly in the revised manuscript as all the references of the article have been updated.

  1. Abbreviations are not appropriately defined throughout the manuscript. I recommend preparing the list with abbreviations in the revised version of the ms.

According to the reviewer’s comment, the abbreviations were double checked throughout the manuscript. Please see adaptations directly in the revised manuscript.

  1. The quality of the figures is quite low. Authors should include the high-resolution pictures in the revised version of ms.

According to the reviewer’s comment, the quality of the figures was improved, and the high-resolution version of the figures were included in the revised manuscript.

  1. The Authors should run spell check and carefully check for typos.

According to the reviewer’s comment, the English language was double checked, and minor spelling issues were carefully checked throughout the manuscript. Please see adaptations directly in the revised manuscript.

Reviewer 3 Report

This is an interesting Review on the involvement of DYRK1A in human diseases.

However, I suggest the following changes.

Major.

1.

3. DYRK1A expression and enzymatic activity.

The authors should also discuss:

-          Kinase TISSUE DISTRIBUTION

-          Kinase substrates specificity (among all the members of the family)

-          Discuss more in-depth Table 1. (substrates identified in vitro or in vivo? subcellular localization of substrates? and go analysis of DYRK1A protein substrates? how many? etc...)

2.

Figure 3. The therapeutic targeting of DYRK1a could require its inhibition or its activation. This does not emerge from Figure 3 where it seems that kinase inhibition is always the best solution. Please indicate in which pathology is required its inhibition or its activation.

3.

DYRK1a targeting should be discussed in-depth and not only in the Conclusion section. Please add a Chapter describing the most relevant DYRK1A inhibitors and a summary table of the most promising available inhibitors.

Minor.

Lane 41. "With more than 84 articles....

substitute with "with 80-90 articles.....

Lane 51. "From insects to humans, and recently also fish, DYRKs are highly
conserved" Please rewrite.

Lane 80. DYRK1A is perhaps the most studied member of the DYRK family.

This is a repeat.

Lane 137. DYRK1A acts on a multitude of exogenous protein substrates.

What does it mean "exogenous"??

Author Response

We thank all the reviewers for the time and hard work dedicated reviewing our article. We have endeavored to take each comment and suggested improvement into account in the accompanying revision. Below we summarize changes made in response to each comment. Please see all the adaptations made in the revised manuscript.

Reviewer 3

  1. This is an interesting Review on the involvement of DYRK1A in human diseases. However, I suggest the following changes.

Major

DYRK1A expression and enzymatic activity. The authors should also discuss:

- Kinase TISSUE DISTRIBUTION

We thank reviewers #3 for this suggestion also suggested by reviewer #1. We have carefully taken all these comments into consideration, and we have adapted this section based on these suggestions. Please see the modifications made in the revised manuscript as follow (p. 8 lines 91-123):

“From kidneys to bone marrow, DYRK1A expression is ubiquitously observed in all human tissues from early embryonic development to adulthood [72,73]. High expression of Dyrk1A was detected in several areas of the adult hindbrain, particularly in the cerebellum and functionally related structures such as the precerebellar and cerebellar nuclei and the vestibular nuclear complex [74]. Subcellular localization of the kinase, however, is still the focus of numerous studies as, in addition to a PEST sequence at its C-terminus (Figure 2), Dyrk1A has a nuclear targeting sequence at its N-terminus. However, studies showed that a substantial amount of endogenous Dyrk1A is localized within the cytoplasm of the cells in the brains of humans, mice, and chickens [75,76]. This dual localization is due to the fact that DYRK1A substrates are both nuclear and cytosolic proteins [10,76] While DYRK1A is mainly localized in the cytosol in endogenous conditions, it is also accumulated in the nucleus when exogenously overexpressed from substrates that shuttle to and from the nucleus [53,58,66,76].

Like most other kinases, DYRK1A depends on a molecular switch to adopt an active/inactive conformation. Whereas the dual-phosphorylation of the MAPKs is a classical paradigm for the on⁄off regulation by upstream protein kinases, DYRKs rely on auto-phosphorylation of an absolutely conserved tyrosine (Y321 for DYRK1A) residue in the activation loop (see Figure 2) and appears to be an evolutionarily conserved ancestral feature 61. It was shown that, for mammalian DYRK1A, tyrosine autophosphorylation is an intrinsic capacity of the catalytic domain and does not depend on other domains or any cofactor. However, Y321 substitution with phenylalanine significantly reduces, at least in vitro, the catalytic activity of DYRK1A and, surprisingly, dephosphorylation does not inactivate mature DYRK1A [22,77]. It is therefore assumed that tyrosine phosphorylation is only required for activation but not for maintenance of the active state.

Based on these findings, DYRK1A has multiple biological functions: signaling, mRNA splicing, chromatin transcription, DNA damage repair, cell survival, cell cycle control, neuronal development and functions, synaptic plasticity, etc. [10–12,19,78]. The level of activity of DYRK1A is of key importance for its physiological and pathological effects [24,79,80]. Hence, DYRK1A is regulated based on both gene expression and protein abundance (for reviews see [77–79,81]) with both low and high expression exerting significant effects on human diseases (Table 1). Due to its low tissue specificity, multiple proteins are targeted by DYRK1A overexpression, leading to a multitude of various changes within the cell and, consequently, to a plethora of different symptoms and pathologies”.

  1. Kinase substrates specificity (among all the members of the family). Discuss more in-depth Table 1. (substrates identified in vitro or in vivo? subcellular localization of substrates? and go analysis of DYRK1A protein substrates? how many? etc...)

According to reviewers #1 and #3, the Table 1 has been updated and extensively revised to discuss more into details the targets of DYRK1A. We have modified Table 1 in accordance with this remark. Please see the modification directly in the revised manuscript as many modifications have been made in the revised Table 1.

  1. Figure 3. The therapeutic targeting of DYRK1a could require its inhibition or its activation. This does not emerge from Figure 3 where it seems that kinase inhibition is always the best solution. Please indicate in which pathology is required its inhibition or its activation.

Indeed, both solutions were not clearly presented in Figure 3. We have adapted Figure 3 according to this. Please see the modification made on Figure 3 directly in the revised manuscript.

  1. DYRK1a targeting should be discussed in-depth and not only in the Conclusion section. Please add a Chapter describing the most relevant DYRK1A inhibitors and a summary table of the most promising available inhibitors.

We thank reviewer #3 for this understandable comment. Indeed, DYRK1A targeting was not discussed in-depth as this was not the goal of this review. Not mentioning it would be a mistake, but we did not wanted to give it more emphasis because it would make the reading more laborious and confuse the reader. Our vision focuses solely on the biochemistry of DYRK1A and its involvement and omnipresence in various diseases. We think that a review of DYRK1A a inhibitors is a separate review unto its own. Many reviews are available in the literature on this subject, and we have mentioned them in our references so that the reader can go and consult them if he desires.

Nevertheless, based on this comment, we went through the literature to add all the newly released references on the subject. Please see the adaptation directly in the revised manuscript as all the references of the article have been updated.

Minor

  1. Lane 41. "With more than 84 articles.... substitute with "with 80-90 articles.....

The manuscript has been revised according to the reviewer’s remark. Please see adaptations as follow (p.1 lines 41-43):

“With 80-90 articles published daily, it is clear for the scientific community that this protein superfamily plays a key role in human health, and recently as therapeutic targets for novel drug candidates [8].

  1. Lane 51. "From insects to humans, and recently also fish, DYRKs are highly conserved" Please rewrite.

We thank reviewer #1 and #3 for noticing that this sentence could be misunderstood. We have adapted the sentence based on these suggestions. Please see the modification as follow (p.2 lines 54-56):

“From yeast to humans, and recently also discovered in fish, DYRKs are highly conserved, as orthologous genes have been cloned independently in various eukaryotic organisms (Figure 1) [10,12,13].”

  1. Lane 80. DYRK1A is perhaps the most studied member of the DYRK family. This is a repeat.

We thank reviewer #3 for noticing this repetition. The sentence was adapted to avoid that. Please see below the adaptation in the revised manuscript (p.3 lines 85-86):

“DYRK1A, being the most studied member of the DYRK family, has been implicated in a large and growing number of human diseases.”

  1. Lane 137. DYRK1A acts on a multitude of exogenous protein substrates. What does it mean "exogenous"??

By “exogenous”, we wanted to indicate that the substrates could be of external origin to the cell nucleus, thus demonstrating the wide variety of localization that DYRK1A can target. This issue has been corrected by adding information in Table 1 about the subcellular localization of these substrates as recommended in a previous comment. Please see the adaptation in the revised manuscript and as specified in previous comments.

Round 2

Reviewer 2 Report

The Authors replied properly to my suggestions.

Reviewer 3 Report

The authors asnwer to all my comments.